# Investigating the Direct and Spillover Effects of Urbanization on Energy-Related Carbon Dioxide Emissions in China Using Nighttime Light Data

Li Sun, Xianglai Mao, Lan Feng *, Ming Zhang ⓘ, Xuan Gui and Xiaojun Wu

Hubei Key Laboratory of Regional Ecology and Environmental Change, School of Geography and Information Engineering, China University of Geosciences, Wuhan 430074, China; sunli@cug.edu.cn (L.S.); maoxianglai@cug.edu.cn (X.M.); zhangm@cug.edu.cn (M.Z.); guixuan@cug.edu.cn (X.G.); dd123012@cug.edu.cn (X.W.)
* Correspondence: fenglan@cug.edu.cn

**Abstract:** Cities are the main emission sources of the $CO_2$ produced by energy use around the globe and have a great impact on the variation of climate. Although the implications of urbanization and socioeconomic elements for carbon emission have been extensively explored, previous studies have mostly focused on developed cities, and there is a lack of research into naturally related elements due to the limited data. At present, remote sensing data provide favorable conditions for the study of large-scale and long-time series. Also, the spillover mechanism of urbanization effects on the discharge of carbon has not been fully studied. Therefore, it is necessary to distinguish the types of influence that various urbanization factors have on emissions of $CO_2$. Firstly, this study quantifies the urban $CO_2$ emissions in China by utilizing nighttime lighting images. Then, the spatio-temporal variations and spatial dependence modes of $CO_2$ emissions are explored for 284 cities in China from 2000–2018. Finally, the study further ascertains that multi-dimensional urbanization, socio-economic and climate variables affect the discharge of carbon using spatial regression models. The results indicate that $CO_2$ emissions have a remarkable positive spatial autocorrelation. Urbanization significantly increases $CO_2$ emissions, of which the land urbanization contribution towards $CO_2$ emissions is the most important in terms of spillover effects. Specifically, the data on urbanization's direct effects reveal that $CO_2$ emissions will increase 0.066%when the urbanization level of a city rises 1%, while the spillover effect indicates that an 0.492% emissions increase is associated with a 1% rise of bordering cities' average urbanization level. As for the socio-economic factors, population density suppresses $CO_2$ emissions, while technological levels boost $CO_2$ emissions. The natural control factors effect a remarkable impact on $CO_2$ emissions by adjusting energy consumption. This study can provide evidence for regional joint prevention in urban energy conservation, emission reduction, and climate change mitigation.

**Keywords:** $CO_2$ emissions; urbanization; spatial regression model; spillover effect; meteorological factors

## 1. Introduction

The increase in urbanization has not just entailed great economic development, but also a growing number of environmental issues in China. As the centers of social and economic activities, cities are playing an essential role in the change of the global climate, such as the global warming resulting from the rise of $CO_2$ levels [1], which is linked to a rise in $CO_2$ levels in the atmosphere [2]. Although cities account for only 3% of the land area on earth, they contribute 60–80% and 71% of the global consumption of energy and global emissions of GHGs, respectively, the latter of which is expected to increase to 76% by 2030. Hence, the perpetration of emission reduction strategies in cities is important to the pursuit of low-carbon development and further addressing climate change on a global' scale [3].

China, as the world's largest producer of $CO_2$, is highly vulnerable to the effects that climate change has to offer [4], and has taken several initiatives to reduce emissions of $CO_2$ [5]. In 2008, the Kyoto Protocol entered into force, emphasizing the developing countries' role in decreasing $CO_2$ emissions [6]. According to the 2015 Paris Climate Change Agreement, China needs to reach peak emissions as late as 2030 and carbon neutrality as late as 2060 [7]. During this process, the Chinese government has been facing a dilemma between national economic development and emission reduction. This is not just China's most urgent and critical responsibility, but the world's most urgent challenge [8]. The Chinese government is committed to realizing a higher emission reduction target of 18% when the urbanization rate is increased to 65%. To accurately predict $CO_2$ emissions and design policies to save energy and minimize emissions, an examination of $CO_2$ emission patterns and sources in the past is necessary [5]. This also has important reference significance for China's attainment of the "double carbon" policy objective [9].

Cities are the basic administrative units for China for implementing strategies to mitigate $CO_2$ emissions. Establishing a credible city-level $CO_2$ emissions appraisal is essential for building effective carbon reduction policies [10]. Nevertheless, most of these data rely on statistics released by China's national and provincial statistical offices. Owing to a paucity of statistics on energy consumption in most cities in China, especially in less developed areas, $CO_2$ emissions received little attention in prior investigations at the urban level. The small number of city-scale studies have only focused on some developed cities with complete data. For instance, Wang et al. [11] conducted a study on the comprehensive effects that socio-economic and geographical management variables have on $CO_2$ emissions in five expanded Chinese metropolises. Xu et al. [12] calculated the $CO_2$ emitted by fossil fuel burning and activities of industry in 18 towns in China and found that they contributed 13% of $CO_2$ emissions during the period 2000–2014. Tan et al. [13] used China's provincial panel data to explore the driving factors of $CO_2$ emissions under different urbanization development rates, in order to understand the relationship between urbanization and $CO_2$ emissions and its impact mechanism. To break through this dilemma, some investigators attempted to use night light data from the National Defense Meteorological Satellite Program's operational line scanning system (DMSP-OLS) and the Visible Infrared Imaging Radiometer Suite (VIIRS) on Suomi National Polar Partnership satellites to assess $CO_2$ emissions, as such data can directly indicate the intimate association between $CO_2$ discharge and human activity intensity. In previous research, this approach has been widely used. For example, Wang and Liu [5] used panel data models to assess the decisive factors and spatio-temporal variability regarding $CO_2$ emissions at the city level in China, supporting the traditional environmental Kuznets curve (EKC) theory. Other investigations have reached similar conclusions using the same dataset and further confirmed the EKC hypothesis. Wu et al. [14] used 41 cities in southwestern China as an example to quantify the relationship between urban expansion and carbon emissions using nighttime lighting data. The results showed that urban expansion in small cities had the strongest impact on carbon emissions. Therefore, nighttime light (NTL) data is an available resource for predicting regional $CO_2$ emissions.

Understanding the spatio-temporal characteristics and causes with respect to carbon emissions is becoming a trendy research topic. Su et al. [15] assessed the $CO_2$ emitted by cities in China from 1992 to 2010 and found that their changes were consistent with economic development. High-$CO_2$-emitting cities are located predominantly in the southern and eastern coastal regions of China, while low $CO_2$-emitting cities are positioned within the southwest. Chen and Yang [16] also discovered larger increases in $CO_2$ emissions in eastern provinces relative to central and western China. In recent years, scholars have performed a substantial study of the variables that impact $CO_2$ emissions [17]. The drivers of $CO_2$ emissions have been studied using a variety of factors and quantitative methodologies, including population, economic growth, energy intensity, industrial structure, foreign trade, etc., which are intimately associated with $CO_2$ emissions [17,18]. One of the key difficulties facing the world in the 21st century is urbanization, but the link between urban

expansion and greenhouse gas emissions is controversial. According to one viewpoint, urbanization obstructs the achievement of carbon reduction aims by increasing energy demand. Another viewpoint is that through fostering energy efficiency improvements, urbanization contributes to the attainment of emission reduction objectives and the alleviating of environmental pressures. In addition, due to the distinctive and diverse nature of the urbanization of China, it is vital to investigate the various positions among several aspects of urbanization to fully lead the implementation of urban emission reduction strategies, an area which has evinced a lack of systematic research in previous studies. Consequently, more research into the links between the factors influencing$CO_2$ emissions is necessary. Additionally, meteorological conditions are also thought to affect $CO_2$ discharge. Addressing the effect of natural variables is crucial, yet few researchers have accomplished it. Different models can be used to assess the association between factors in $CO_2$ discharge, and some these indicators have already been developed, including the input–output analysis model (IOA), exponential decomposition analysis model (IDA), general econometrics model, etc. [19]; however, few studies describe and explain the correlations of $CO_2$ emissions in space. As $CO_2$ easily exceeds regional boundaries, it is critical to comprehend the spatial correlation as well as the agglomeration effect of $CO_2$ emitted from adjoining areas, which will help to control cross-pollution between adjacent regions.

To sum up, although previous studies have made effective insights into the connection between urbanization and $CO_2$ emissions, there are still some deficiencies. China covers a large area and has many cities with different sizes, different geographical situations, and development conditions are quite different. The externalities of carbon emission distribution may partly come from the spillover effects of urbanization, that is, the influence of urbanization in neighboring cities. In addition, different urbanization dimensions and the impact of socio-economic factors brought about by urbanization need to be further explored, and the impact of climate factors is rarely included in a study dealing with energetic-related carbon emission in cities. Accordingly, given the above deficiencies, this paper sheds light on the influence of urbanization on carbon emissions by quantifying Chinese cities' $CO_2$ emissions. The specific research objectives are as follows: (1) to quantify Chinese cities' $CO_2$ emissions and reveal their spatio-temporal characteristics and spatial dependence; (2) to quantitatively evaluate the spillover effects and direct effects from urbanization on carbon emissions; and (3) to comprehensively examine the action mechanisms of various factors of carbon emissions. To meet the above goals, based on a reasonable quantification of carbon emissions from cities in China, 284 cities will be studied in this essay, and a spatial regression model will be employed to comprehensively examine the effect of urbanization on carbon emissions. This study can provide a logical and effective method which can be used to understand the association that exists between carbon emissions and urbanization and present some reasonable ways for China to effectively implement $CO_2$ emission reduction measures.

## 2. Materials and Methods

### 2.1. Estimation of City-Scale $CO_2$ Emissions in China

Since there is a serious shortage of urban-level "energy consumption" statistics relevant to the discharge of $CO_2$ in China, the energy-related emissions of $CO_2$ from Chinese cities are estimated by using globally harmonized NTL images in this research.

Firstly, we calculate the discharge of $CO_2$ for provinces and cities that have complete energy statistics utilizing the basic guidelines and uniform standards for assessing international greenhouse gas $CO_2$ emissions, as provided by the IPCC guidelines [20]. For this, data on energy consumption by various provinces and cities were obtained from the *China Energy Statistical Yearbook* and the energy balance tables of specific cities. The emissions of carbon are measured by:

$$CO_2 = \frac{44}{12} \sum_{i=1}^{9} K_i E_i \qquad (1)$$

where $i$ stands for different forms of energy, like crude oil, coke, raw coal, and so on. $E_i$ refers to the standard coal usage of category $i$ energy. $K_i$ represents the emissions factor of carbon for energy source category $i$. The standard coal conversion factor and emissions factor of $CO_2$ for those various sources of energy are set out in Table 1.

**Table 1.** Calculation coefficients of various energy sources.

| Energy Type | Raw Coal | Coke | Crude Oil | Gasoline | Kerosene | Diesel Oil | Fuel Oil | Natural Gas | Heat | Electricity |
|---|---|---|---|---|---|---|---|---|---|---|
| Standard coal factor tce/t | 0.7143 | 0.9714 | 1.4286 | 1.4714 | 1.4714 | 1.4571 | 1.4286 | 1.33 | 34.12 | 0.1229 |
| Carbon emission factor tC/tce | 0.7559 | 0.8550 | 0.5857 | 0.5538 | 0.5714 | 0.5921 | 0.6185 | 0.4483 | 0.67 | / |

Note: The tcesignifies 1 t of coal-equivalent.

Secondly, we use China's annual land cover dataset throughout 2000–2018 to extract impervious areas. This dataset is derived from 335,709 images of Landsat data on Google Earth Engine, with a spatial resolution of 30 m, and the accuracy of visual interpretation is 80% [21]. After that, using the consistent NTL data generated by coordinating the calibrated DMSP and VIIRS data, the total night-lighting value of each province and city is calculated based on impervious areas, respectively [22]. A reliable linear relationship (R = 0.946, $p < 0.001$) was found between the statistical $CO_2$ emissions and the overall digital numbers (DN) from night lights (Figure 1), which has also been verified in previous studies. Thus, we can feasibly calculate carbon emissions at an urban scale in China.

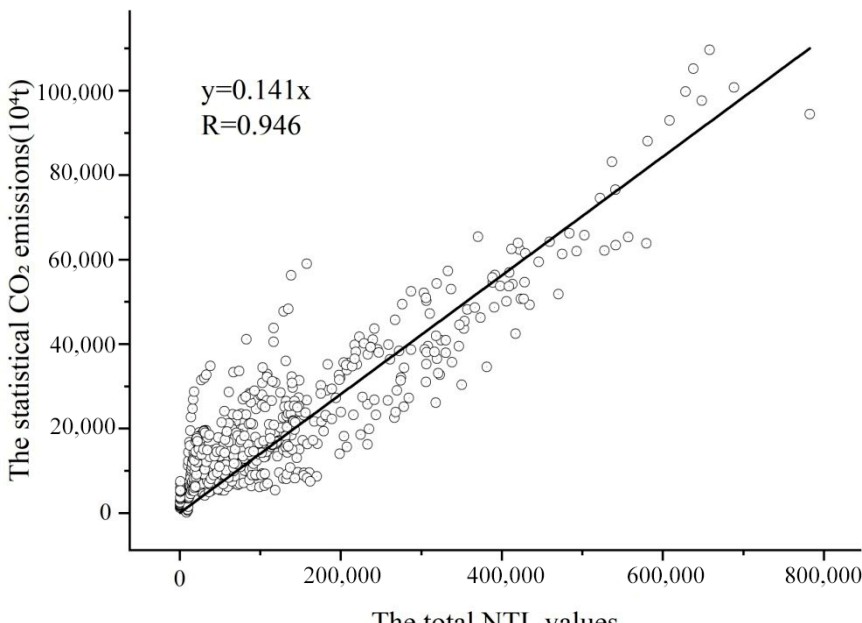

**Figure 1.** The relationship between the total nighttime light values and $CO_2$ emissions. The total NTL is the sum of the nighttime light pixel values in a city area.

### 2.2. Urbanization and Meteorological Factors

This paper studies several urbanization aspects, including population urbanization (PU), economic urbanization (EU) and land urbanization (LU). The proportion of the nonagricultural population is represented by PU; EU is expressed by the GDP ratio of secondary industries; LU is a metric measuring how much built-up area there is in a city. To fully exploit the effect of urbanization in terms of the emission of $CO_2$, we finally selected the following nine variables to be included in our study, as they are widely used

for describing the socio-economic changes brought about by urbanization. The following is a description of them:

(1) Population Density (PD), expressed per unit of population in a city, is a common research indicator. A denser population shows stimulated consumption and further increases in production activities. Furthermore, residents' production and consumption activities will lead to greater pollution emissions. As a result, the economies of scale caused by increased population density may lead to more $CO_2$ emissions. Simultaneously, the sharing of technology and knowledge brought about by population aggregation is more productive than are isolated individuals; thus, it greatly improves production efficiency [23]. The agglomeration effect of the population can reduce the per capita discharge of $CO_2$. Accordingly, the consequences of PD for the discharge of $CO_2$ remain uncertain.

(2) Per capita GDP (PGDP) serves as the measure for a city's economic progress. A greater degree of growth of the economy tends to increase drastically the amount of energy consumed, which results in an increase in output [24]. However, economic development, to a certain extent, will help to improve people's awareness of environmental protection, which is conducive to emission reduction [25]. Consequently, the PGDP result for the discharge of $CO_2$ is puzzling.

(3) Industrial Structure (IS) is determined by means of the percentages of GDP produced by primary, secondary and tertiary industries. The secondary industry, which is dominated by industrial production, often causes higher levels of consumption in terms of energy, as well as the discharge of pollution emissions. The concept of tertiary sector is adopted to describe industrial optimization and upgrading. The optimization and upgrading of IS may enhance energy performance, stimulating the utilization of cleaning energy and optimizing the distribution within social resources, which is seen as an efficacious way to reduce the intensity of the discharge of carbon [26]. When IS is optimized, the conventional energy-intensive industries are substituted for by high-tech industries, which will save the usage of energy and affect the discharge of carbon [27,28]. For that reason, this variable holds the promise of curbing $CO_2$ emissions.

(4) Foreign Direct Investment (FDI) stands as a proxy for the extent of trade in foreign areas, which is thought to be among the important factors governing $CO_2$ emissions [29]. And yet, it is matter of controversy whether the impact of foreign trade will bring greater economic benefits or more severe environmental problems. In one respect, according to the "pollution paradise" hypothesis, foreign high-energy-consuming enterprises always invest in nations with inadequate environmental restrictions to escape high environmental costs. Foreign-trade-driven industrial prosperity often requires more energy consumption, so FDI increases the $CO_2$ emissions of host countries. In another aspect, the "pollution halo" hypothesis shows that, due to the demonstration effect of environmentally friendly technology, multinational corporations with higher requirements for environmental protection appear to benefit the host country's environment, promoting advances in technology and growth of economic strength, and thus restraining the discharge of carbon [30,31]. Therefore, the FDI's impact on $CO_2$ emissions continues to be ambiguous.

(5) Government Intervention (GI) is determined as the general government expenditure as a percentage of GDP. The policy adjustments of government departments can maximize social welfare [32], and this is an indispensable part of addressing resource depletion and environmental issues [33]. The effective realization of emission reduction strategies requires government departments to give full play to their roles. Therefore, this variable is projected to reduce carbon emissions.

(6) Technical Level (TL) is measured by the percentage of research and technology spending relative to the aggregate financial outlay. TL is a dual-edged sword in the fight against China's carbon emissions crisis [34]. First, technological advancement is among the most suitable ways to cut carbon emissions, preserve resources and promote economic development through the implementation of energy-saving and sus-

tainable business practices. Others contend that, when innovation efforts are primarily directed towards increasing the productivity of conventional factors, technological progress would increase pollutant emissions as a result of the spread of mass production, which would result in increased production and pollution and increased carbon emissions [34]. Consequently, the impact on the discharge of carbon from the level of technology is likewise unknown.

(7) Energy Intensity (EI) is expressed in terms of total electricity consumption. The intensity of energy has been already documented as a key driver affecting carbon emissions [35], and electricity consumption stimulates the discharge of $CO_2$ [36]. Therefore, emissions of carbon are projected to benefit from this variable.

(8) Traffic congestion (TC) is expressed in terms of surfaced road area per capita. One of the primary causes of carbon emissions that contribute to the greenhouse effect is road traffic [37]. Traffic jams will reduce the fuel economy of vehicles, causing wasteful energy and inordinate discharge of $CO_2$. In addition, convenient transportation will increase the demand for vehicles and may also promote $CO_2$ emissions [38,39]. Accordingly, the impact on the discharge of carbon from TC remains uncertain.

(9) Public transport (PT) is expressed in terms of bus ownership per ten-thousand people. As part of durable municipal design, the potential of mass transit to reduce emissions was studied. The development of buses, among the important ways to limit the usage of private cars that provide public transportation [40], can lower $CO_2$ emissions and energy usage and improve road safety. Thus, this factor should cut $CO_2$ emissions.

In addition to socio-economic factors, carbon emissions will also be affected by meteorological factors. Stable meteorological conditions can inhibit the dispersion of atmospheric pollutants, and meteorological factors often affect energy consumption, affecting carbon emissions. Within this study, four meteorological indices are selected, namely, precipitation rate (PREC), downward shortwave radiation (SRAD), near-surface air temperature (TEMP), and near-surface total wind speed (WIND), which are inserted in the model to diminish the bias resulting from missing variables.

Due to data limitations, the time span of this study is 2000–2018. All yearly social and economic data during 2000–2018 were obtained from China's urban statistical yearbook and national economic and social development statistical bulletin. The meteorological data are derived from the surface meteorological elements dataset in China issued through the National Qinghai–Tibet Plateau Scientific Data Center. This is China's first high-spatio-temporal-resolution grid near-surface meteorological dataset specializing in the research on land surface patterns, with spatial-temporal resolutions of 0.1° and 3 h, respectively. See Table 2 for all variables, with descriptive statistics.

**Table 2.** Description of statistics variables used in this study.

| Variables | N | Mean | s.d. | Min | Max |
|---|---|---|---|---|---|
| PU | 5396 | 37.035 | 19.139 | 7.136 | 100.000 |
| LU | 5396 | 1.415 | 3.166 | 0.011 | 46.470 |
| PD | 5396 | 424.460 | 376.480 | 4.547 | 11,563.707 |
| PGDP | 5396 | 34,115.502 | 42,868.909 | 603.991 | 532,702.377 |
| EU | 5396 | 47.270 | 11.207 | 13.520 | 90.970 |
| IS | 5396 | 37.556 | 8.890 | 8.500 | 80.980 |
| FDI | 5396 | 63,201.465 | 175,640.958 | 1.000 | 3,082,563.000 |
| GI | 5396 | 14.776 | 10.543 | 0.499 | 91.551 |
| TL | 5396 | 1.127 | 1.377 | 0.004 | 19.162 |
| EI | 5396 | 804,344.902 | 1,409,824.958 | 4527.000 | 15,666,595.000 |
| TC | 5396 | 3.770 | 5.548 | 0.004 | 84.213 |
| PT | 5396 | 2.948 | 6.411 | 0.031 | 115.006 |
| $CO_2$ | 5396 | 2144.918 | 2678.390 | 7.614 | 24,053.895 |
| PREC | 5396 | 0.118 | 0.063 | 0.008 | 0.365 |
| SRAD | 5396 | 157.066 | 17.469 | 107.643 | 223.191 |
| TEMP | 5396 | 287.087 | 5.438 | 270.726 | 298.750 |
| WIND | 5396 | 2.267 | 0.684 | 0.777 | 5.977 |
| Number | 284 | 284 | 284 | 284 | 284 |

Note: N is the number of datapoints, mean is the average of the variable, s.d. is the standard deviation, and min and max represent the minimum and maximum values of variables, respectively.

*2.3. Methods Used*

2.3.1. Temporal Trend

The slope of $CO_2$ emissions is calculated by a univariate linear regression model for time trend analysis. Through the linear relationship between time and emissions of carbon for each city, the slope can capture the changing trend [41], and it is worked out by the below equation:

$$Slope = \frac{\sum_{i=1}^{n} i \cdot Y_i - \frac{1}{n} \left(\sum_{i=1}^{n} i\right)\left(\sum_{i=1}^{n} Y\right)}{\sum_{i=1}^{n} i^2 - \frac{1}{n}\left(\sum_{i=1}^{n} i\right)^2} \tag{2}$$

where $Y$ represents the carbon emissions of every unit of study, and $n$ represents the period, while $i$ represents the year. If the $Slope > 0$, it indicates that the property increases over time. Otherwise, it is the opposite. The absolute slope captures the rate at which it increases or decreases.

2.3.2. Spatial Dependence Analysis

Understanding the spatial autocorrelation of the discharge of carbon becomes critical for explaining the overall spatial spillover impact of urbanization upon the discharge of $CO_2$. Moran's I index was used to examine the spatial autocorrelation of the discharge of $CO_2$ in this article. The Moran index for the entire region was determined by using the below equation:

$$I = \frac{\sum_{i=1}^{n} \sum_{j \neq i}^{n} (x_i - \overline{x})(x_j - \overline{x})}{S^2 \sum_{i=1}^{n} \sum_{j=1}^{n} W_{ij}} \tag{3}$$

where $S^2 = (1/n)\sum_{i=1}^{n}(x_i - \overline{x})^2$ and $\overline{x} = (1/n)\sum_{i=1}^{n} x_i$. $x_i$ and $x_j$ are the observed values of factorial variables in region $i$ and region $j$, respectively; $n$ is the number of sample regions and $W_{ij}$ is the spatial weight matrix element. The $I$ belongs to $[-1, 1]$, which is a higher absolute number, the more spatial autocorrelation there is [42]. Moran's $I$ is transformed to Z-scores like in the below:

$$Z = \frac{I - E(I)}{\sqrt{VAR(I)}} \tag{4}$$

where $E(I)$ is just the desired value of said $VAR(I)$ and Moran's $I$ is its variance. $Z > 1.96$ shows there is a positive spatial autocorrelation among the discharge levels of carbon of each region and $Z < -1.96$ implies a spatial negative autocorrelation among the emission levels of carbon of each region. In addition, the value range of $Z$ of $[-1.96, 1.96]$ implies that no spatial autocorrelation exists. Lastly, we also use Getis-OrdGi* statistics provided by ArcGIS10.3 to determine hot and cold carbon emissions with statistically significant results.

2.3.3. Spatial Regression Analysis

As the major source of the greenhouse effect and carbon emissions, $CO_2$ has significant mobility; coupled with the geographical relationship between regions, $CO_2$ emissions will be affected not just by the city's own conditions, but also by those of the adjacent cities. Empirical analysis conducted based on the viewpoint of spatial spillover effects can describe the actual spatial effects of the discharge of $CO_2$ more completely and accurately. The classic ordinary least squares (OLS) model disregards spatial dependence, which might cause the estimated results to be inconsistent with the actual situation; this can be well-addressed with the spatial econometric model [43,44]. At present, the major econometric models used to examine spatial impacts are the spatial lag model (SLM), spatial error model (SEM), and spatial Durbin model (SDM). The formulae are as follows:

$$SLM : y = \rho Wy + X\beta + \varepsilon, \varepsilon \sim \left(0, \sigma^2 I_n\right) \tag{5}$$

$$SEM : y = X\beta + \varepsilon, \varepsilon = \delta W\varepsilon + e \tag{6}$$

$$SDM : y = \rho Wy + X\beta + \theta WX + \varepsilon \tag{7}$$

where $y$ is the dependent variable, $X$ is the independent variable, $W$ is the spatial weight matrix, $\rho$ is the spatial autoregressive coefficient, $\beta$ and $\theta$ are the regression coefficients, and $\varepsilon$ is the residual term [45].

For the above three models, we need to pass the following tests so that we can make a choice. First, the panel data are diagnosed by the Lagrange multiplier (LM) test, which demonstrates the rationality of introducing a spatial econometric model. Secondly, SLM, SEM and SDM are screened based on the likelihood ratio (LR) test, while the model's individual and time effects are tested. Finally, the Hausman test is used for selecting the fixed-effects or random-effects model to obtain the appropriate spatial effects measurement model [46,47].

## 3. Results

### 3.1. Spatio-Temporal Features of $CO_2$ Emissions from Chinese Cities

To further reveal the impact that urbanization has made on the discharge of $CO_2$, a spatial econometric model is applied for evaluating the influence mechanism of urbanization on the discharge of $CO_2$. This study estimates the carbon emission levels from Chinese cities during 2000–2018 with the use of NTL images (Figure 2). The characteristic pattern of the regional $CO_2$ emission spatial distributions for Chinese cities is "more in the east and less in the west" as a whole. According to the results of the $CO_2$ emissions spatial distribution for 2000–2018, there is an overall growth trend, and high-emission areas represented by the BTH spread to surrounding cities.

Because of the defects in some city data, we carried out a follow-up study on the final 284 cities. Combined with the results of Figure 2, the results indicate that Beijing has always been the Chinese metropolis with the highest $CO_2$ emissions, nearly doubling its carbon emissions from 1.256 Mt in 2000 to 2.405 Mt in 2018. In addition to Beijing, there were also high carbon emissions in Tianjin (0.886 Mt) and Shanghai (0.874 Mt) in 2000. Prior to 2018, the carbon emissions of 26 cities, such as Suzhou, Qingdao, Weifang, Zhengzhou and Tangshan, each exceeded 0.8 Mt. In addition, we also observe that the average yearly growth rate of carbon emissions in Baoshan City (68.574%) is the highest, followed by Zunyi City (51.554%) and Luzhou City (50.564%). Although these cities have relatively low total carbon emissions, within the last 19 years, they have also experienced rapid development. China's 284 cities' total $CO_2$ emissions rose from 3.389 billion tons in 2000 to 8.818 billion tons in 2018 (Figure 3). Although the annual growth rate has a slight downward trend, the discharge of $CO_2$ in China is still on the rise. According to the statistical results, the $CO_2$ releases from the selected Chinese cities fall well below the average (Figure 4), indicating that most Chinese cities emit high levels of $CO_2$. In addition, we use the univariate linear regression method to fit the slope of Chinese cities' carbon emissions to show their regional differences (Figure 5). The cities with the fastest growth in $CO_2$ emissions are Suzhou (7.83), Tianjin (7.62), Shanghai (6.24) and Beijing (6.18). Secondly, the carbon emissions of the core cities of the said three primary coastal urban agglomerations are growing rapidly. Furthermore, those inland cities with faster growth trends are mainly capital cities, such as Chengdu, Chongqing and Wuhan. Then we calculate Moran's I for assessing the overall spatial clustering effect for $CO_2$ emissions (Table S1). The results reveal that a strong positive spatial autocorrelation exists among Chinese cities in terms of $CO_2$ emissions from 2000 to 2018.To learn more about the geographic correlation features of the discharge of $CO_2$ for the municipalities in China, we conduct a hot-spot analysis to investigate specific hotspots for carbon emissions (Figure 6). The hotspots of significant carbon emissions are almost all located within BTH, gradually spread to the YRD from 2000 to 2018 and are concentrated in coastal cities. In the early years, Northeast China also had hot cities with carbon emissions, probably because Northeast China's industrial structure is dominated by highly energy-intensive and inefficient heavy industries. The disappearance of carbon emission hotspots in Northeast China after 2010 may be linked to the transformation of the region's industrial structure. The regional distribution of significant cold points of carbon emissions is relatively scattered, occurring mainly in Sichuan, Chongqing and Yunnan. In addition, we observe that there are no hot areas of carbon emissions in the highly urbanized and economically developed Pearl River Delta (PRD) urban agglomeration, which may be related to the differences in $CO_2$ emissions within the surrounding locations.



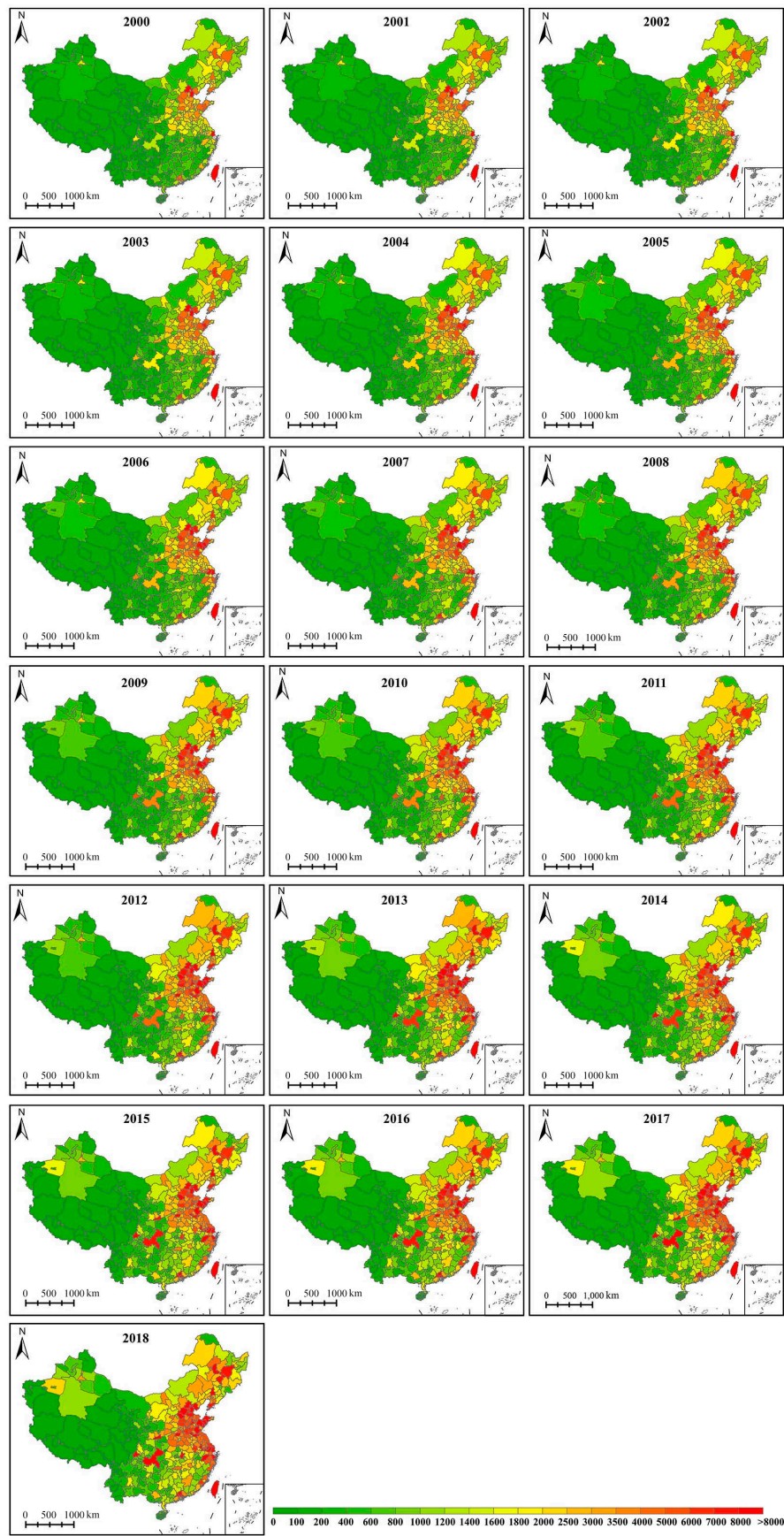

**Figure 2.** Temporal and spatial changes of China's national city-level $CO_2$ emissions (unit: $10^4$ tons), for 2000–2018.

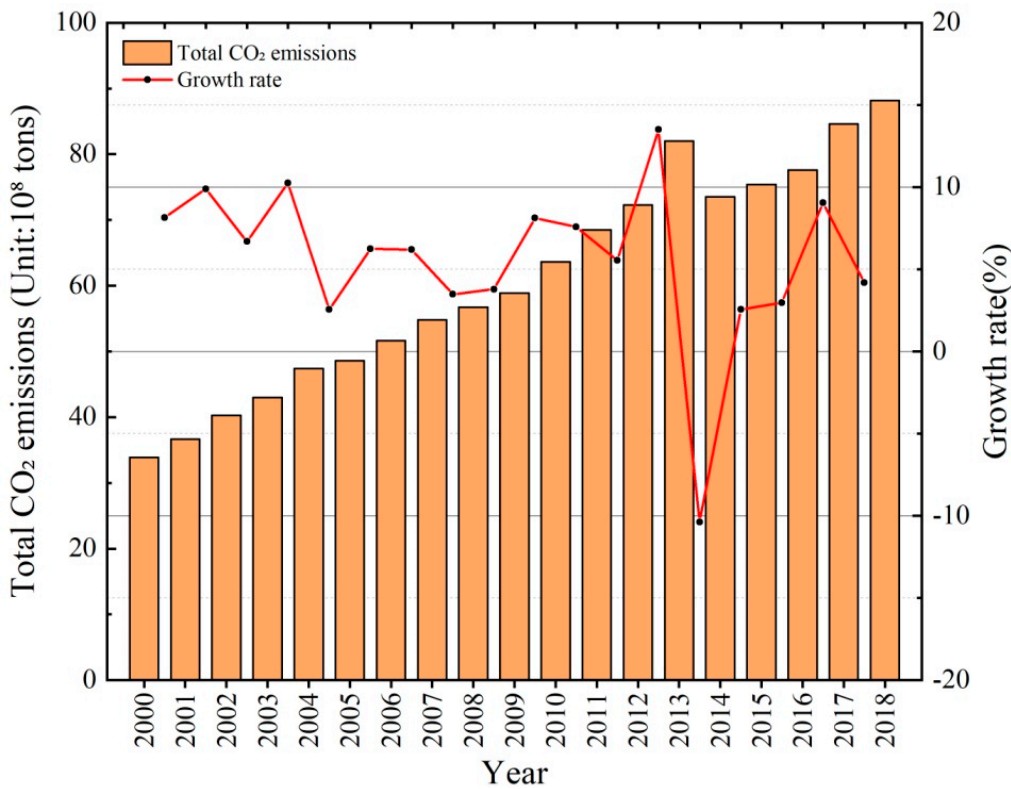

**Figure 3.** Total CO$_2$ emissions and annual growth rate of 284 cities in China during 2000–2018.

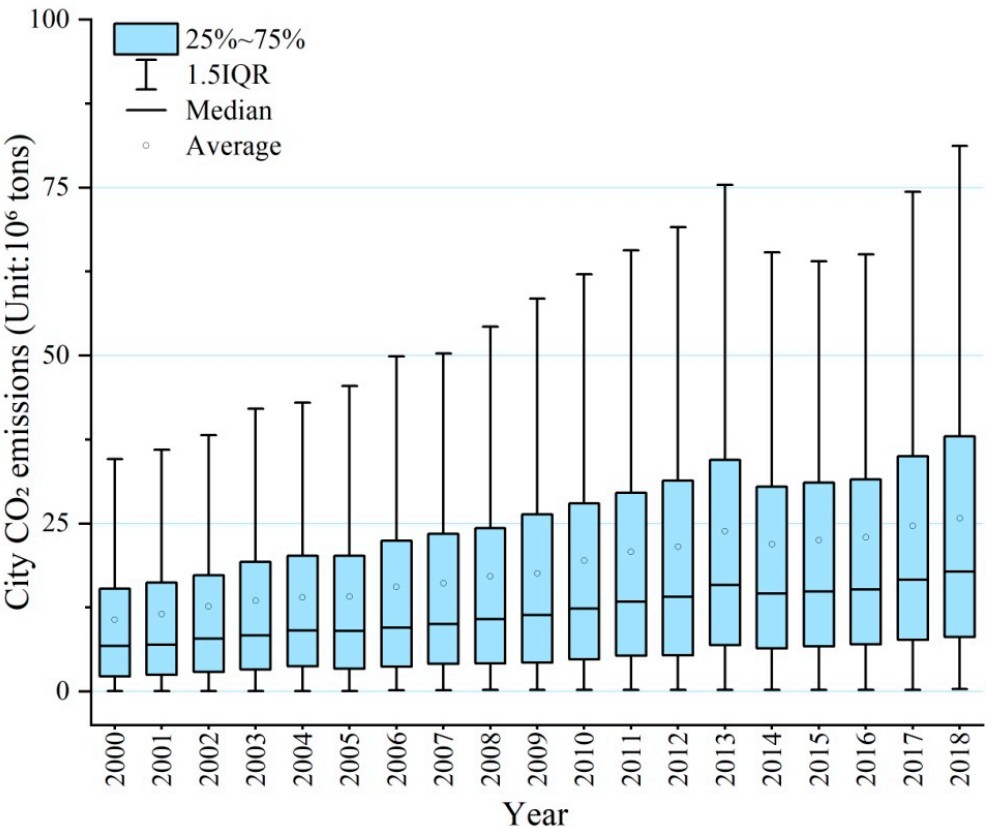

**Figure 4.** Carbon emission statistics of 284 cities in China during 2000–2018.

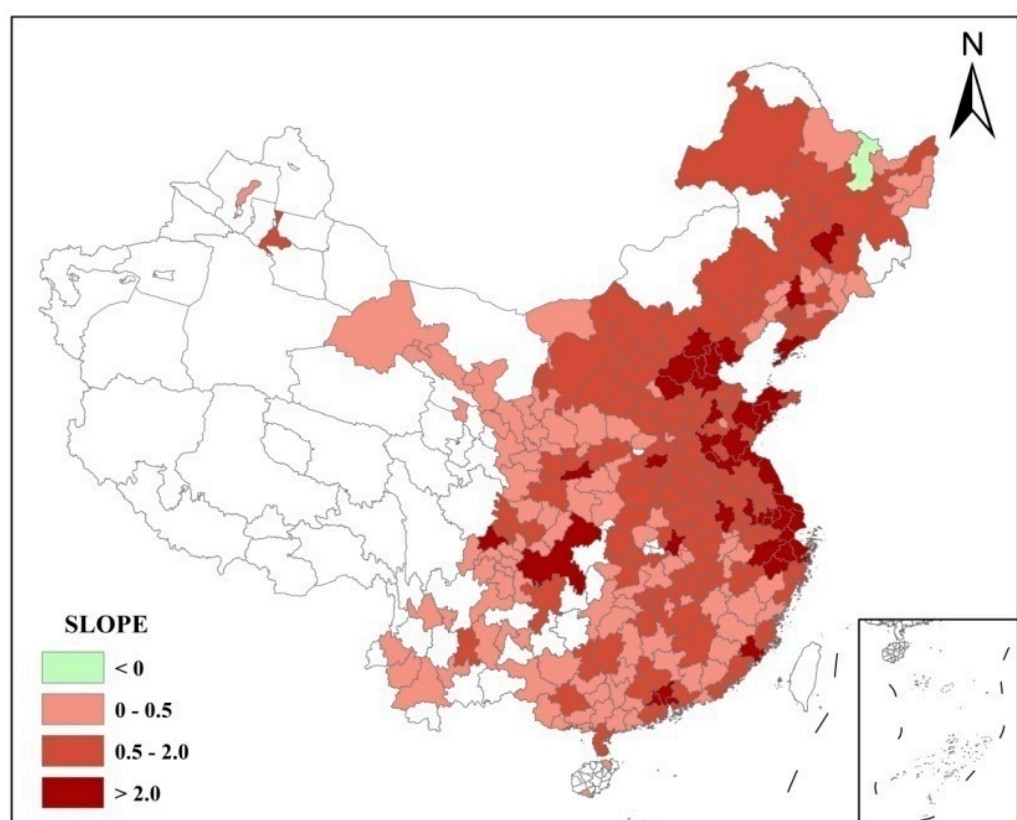

**Figure 5.** Analysis of the change in the trends of $CO_2$ emissions of 284 cities in China from 2000-2018.

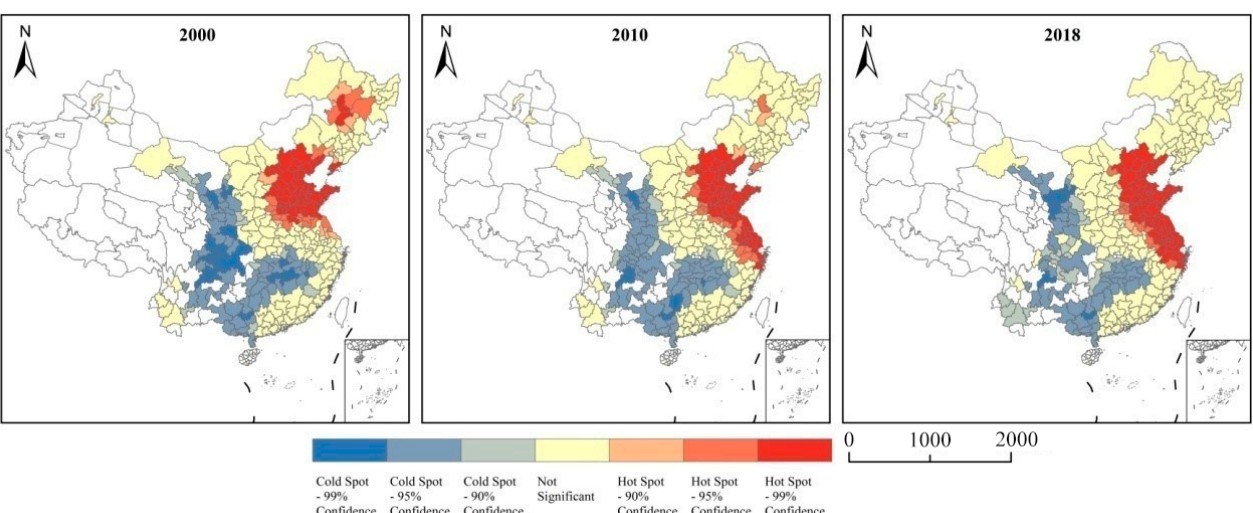

**Figure 6.** Hot and cold spots in the distributions of $CO_2$ emissions of 284 cities in China in 2000, 2010 and 2018.

### 3.2. Direct and Spillover Effects of Urbanization on $CO_2$ Emissions

To fully complete the findings as to these drivers that cause $CO_2$ emissions, firstly we examine the influence levels of urbanization indicators, socio-economic factors and meteorological elements for emissions of $CO_2$ using the OLS model (Table 3). The F-test findings demonstrate that the model appears statistically significant ($p < 0.001$), and the adjusted $R^2$ (0.709) indicates that the model's reliability coefficient is higher. The variables have a variance inflation factor (VIF) below 10, which means the influence of multicollinearity among variables can be ignored. These coefficients show that urbanization seems to be making a major contribution to the discharge of $CO_2$. Population urbanization

affects $CO_2$ discharge positively, while the urbanization of land and increased economic activity are negatively correlated with emissions of carbon. Most of the control variables significantly increase the discharge of $CO_2$. In particular, the strongest impact is seen in PD, with each 1% increase being responsible for a 1.020% increase in the emissions of carbon. Secondly, per capita GDP (0.231), technical level (0.293) and energy intensity (0.389) also have significant favorable effects on $CO_2$. Government intervention and public transport are both significantly inversely connected to $CO_2$ emissions, and each 1% rise will lead to a reduction of 0.254% and 0.181%, respectively. In addition, meteorological factors also have a certain impact on $CO_2$ emissions. A strong positive correlation exists between wind speed and $CO_2$ emissions, while strong negative correlations exist between precipitation, solar radiation and temperature and $CO_2$ emissions.

**Table 3.** Global regression analysis (OLS model).

| | Coefficient | Std. Error | t Value | P (>\|t\|) | VIF |
|---|---|---|---|---|---|
| lnPU | 0.479 *** | 0.333 | 14.36 | 0.000 | 2.49 |
| lnEU | −0.261 *** | 0.062 | −4.24 | 0.000 | 2.41 |
| lnLU | −0.529 *** | 0.300 | −17.60 | 0.000 | 4.71 |
| lnPD | 1.020 *** | 0.322 | 31.72 | 0.000 | 2.07 |
| lnPGDP | 0.231 *** | 0.236 | 9.81 | 0.000 | 4.63 |
| lnTL | 0.293 ** | 0.128 | 2.29 | 0.022 | 1.78 |
| lnIS | −0.001 | 0.660 | −0.01 | 0.990 | 2.79 |
| lnFDI | 0.170 *** | 0.072 | 23.80 | 0.000 | 2.06 |
| lnEI | 0.389 *** | 0.014 | 27.98 | 0.000 | 3.38 |
| lnGI | −0.254 *** | 0.015 | −17.19 | 0.000 | 1.14 |
| lnPT | −0.181 *** | 0.020 | −9.20 | 0.000 | 3.11 |
| lnTC | 0.095 *** | 0.250 | 3.81 | 0.000 | 3.07 |
| lnPREC | −0.677 ** | 0.029 | −2.36 | 0.018 | 2.95 |
| lnSRAD | −0.651 *** | 0.109 | −5.95 | 0.000 | 1.31 |
| lnTEMP | −22.523 *** | 0.990 | −22.74 | 0.000 | 3.54 |
| lnWIND | 0.587 *** | 0.038 | 15.64 | 0.000 | 1.20 |

Note: adjusted R-squared, 0.709; F-statistic, 820.58 on 16 and 5379 DF; *p*-value: 0.000. *** $p < 0.01$, ** $p < 0.05$.

Previous findings indicate a strong spatial autocorrelation of $CO_2$ emissions, and the results of traditional OLS models do not represent the actual impact of each variable well. We therefore choose the model that considers their spatial effect (Table S2). First and foremost, the Lagrange multiplier (LM) test findings all pass the 1% significance test, which points to the presence of spatial lag and spatial error effects. Secondly, the likelihood ratio (LR) experiment findings suggest that SDM is more suitable than SLM and SEM. Finally, the Hausman test findings suggest that a fixed-effects model outperforms a casual one. To sum up, the SDM with fixed effects is chosen for our follow-up study. The $R^2$ also demonstrates that the SDM model is stronger than the OLS model in terms of robustness.

We control for all the socio-economic variables and approach the question of urbanization's impact on emissions of $CO_2$ through the three perspectives of land, economy and population. Table 4 shows how several urbanization aspects affect $CO_2$ emissions under SDM. The order in which different urbanization dimensions have an influence on carbon emissions isas follows: LU > EU > PU. Specifically, PU and EU have significant direct impact coefficients. However, their indirect impact coefficients are insignificant, implying that both PU and EU mainly affect carbon emissions through the local level of urbanization. In addition, the growth of urbanization in neighboring urban areas has a much stronger effect on emissions of $CO_2$than does growth in the local city itself. LU has a direct impact factor of 0.066, indicating that each 1% increase in a city's own urbanization increases its $CO_2$ emissions by 0.066%; the spillover impact coefficient is 0.492, which indicates that each 1% rise in average urbanization rate for adjacent cities increases a city's $CO_2$ emissions by 0.492%.

**Table 4.** Impact of urbanization on $CO_2$ emission in China, based on multiple dimensions of urbanization, from 2000 to 2018.

| Variables | Direct | Indirect | Total | Direct | Indirect | Total | Direct | Indirect | Total |
|---|---|---|---|---|---|---|---|---|---|
| lnPU | 0.039 *** | −0.072 | −0.033 | | | | | | |
| | (3.242) | (−0.957) | (−0.408) | | | | | | |
| lnEU | | | | 0.220 *** | −0.120 | 0.099 | | | |
| | | | | (11.643) | (−1.208) | (0.925) | | | |
| lnLU | | | | | | | 0.066 *** | 0.492 *** | 0.558 *** |
| | | | | | | | (7.906) | (8.447) | (8.770) |
| lnPD | 0.042 ** | −0.313 ** | −0.271 ** | 0.032 * | −0.239 * | −0.207 | −0.027 | −0.786 *** | −0.813 *** |
| | (2.341) | (−2.563) | (−2.039) | (1.813) | (−1.935) | (−1.542) | (−1.374) | (−6.000) | (−5.673) |
| lnPGDP | 0.127 *** | 0.278 *** | 0.405 *** | 0.107 *** | 0.224 *** | 0.330 *** | 0.121 *** | 0.210 *** | 0.331 *** |
| | (13.258) | (4.228) | (5.892) | (11.052) | (3.301) | (4.652) | (12.752) | (3.448) | (5.194) |
| lnTL | 0.026 *** | 0.071 *** | 0.097 *** | 0.028 *** | 0.072 *** | 0.100 *** | 0.024 *** | 0.044 *** | 0.068 *** |
| | (9.339) | (4.026) | (5.177) | (10.084) | (4.035) | (5.252) | (8.534) | (2.694) | (3.878) |
| lnIS | −0.033 ** | −0.329 *** | −0.363 *** | 0.093 *** | −0.341 *** | −0.247 ** | −0.041 *** | −0.365 *** | −0.406 *** |
| | (−2.278) | (−3.828) | (−3.905) | (4.903) | (−3.093) | (−2.060) | (−2.844) | (−4.562) | (−4.699) |
| lnFDI | 0.006 *** | −0.031 *** | −0.026 ** | 0.004 *** | −0.026 ** | −0.022 * | 0.007 *** | −0.018 * | −0.011 |
| | (3.567) | (−3.050) | (−2.304) | (2.736) | (−2.442) | (−1.881) | (4.289) | (−1.898) | (−1.092) |
| lnEI | 0.047 *** | 0.187 *** | 0.234 *** | 0.042 *** | 0.170 *** | 0.212 *** | 0.046 *** | 0.145 *** | 0.190 *** |
| | (10.188) | (5.541) | (6.383) | (9.178) | (4.960) | (5.698) | (9.980) | (4.710) | (5.661) |
| lnGI | 0.024 *** | −0.064 * | −0.040 | 0.016 *** | −0.077 ** | −0.061 | 0.021 *** | −0.095 *** | −0.074 ** |
| | (5.075) | (−1.766) | (−1.041) | (3.352) | (−2.077) | (−1.546) | (4.553) | (−2.811) | (−2.057) |
| lnPT | 0.018 *** | −0.066 | −0.048 | 0.013 ** | −0.063 | −0.049 | 0.010 * | −0.158 *** | −0.148 *** |
| | (3.181) | (−1.511) | (−1.021) | (2.493) | (−1.493) | (−1.085) | (1.769) | (−3.939) | (−3.405) |
| lnTC | 0.031 *** | 0.143 *** | 0.175 *** | 0.030 *** | 0.150 *** | 0.180 *** | 0.020 *** | 0.041 | 0.061 |
| | (5.073) | (2.898) | (3.259) | (4.923) | (2.970) | (3.294) | (3.273) | (0.870) | (1.200) |
| rho | 0.746 *** | | | 0.754 *** | | | 0.726 *** | | |
| | (70.366) | | | (71.354) | | | (65.921) | | |
| sigma2_e | 0.009 *** | | | 0.009 *** | | | 0.009 *** | | |
| | (50.773) | | | (50.703) | | | (50.825) | | |
| Observations | 5396 | | | 5396 | | | 5396 | | |
| R-squared | 0.847 | | | 0.847 | | | 0.862 | | |
| Number | 284 | | | 284 | | | 284 | | |

Note: z-statistics in parentheses. *** $p < 0.01$, ** $p < 0.05$, * $p < 0.1$.

Considering the close relationships between $CO_2$ emissions, population, economy and land urbanization, the comprehensive urbanization index (CU) is determined using three separate levels of urbanization (standardizing each single urbanization index before synthesizing those indicators with the same weight). Table 5 shows the estimated results of CU. Urbanization will significantly increase $CO_2$ emissions, which also demonstrates that increased urbanization impact on emissions of $CO_2$ is mainly affected by land urbanization. As meteorological factors are rarely considered in related research on carbon emissions, we analyze the effect of meteorological factors (Table 5). All meteorological variables pass the test of significance, and the overall model fit is slightly improved. PREC, SRAD, TEMP and WIND all show a remarkable proactive effect on $CO_2$ emissions, particularly temperature. The direct-effect factor of temperature is positive, whereas the indirect-effect factor is negative, indicating that an increase in temperature in a city would raise local discharge of $CO_2$ while lowering discharge of $CO_2$ in adjacent cities. Generally, it is believed that rising population results in higher energy consumption, which will increase $CO_2$ emissions, while improved technology will improve energy efficiency, thereby reducing $CO_2$ emissions. But we have observed an interesting result, in that PD is adversely linked to $CO_2$ emissions, while technological advancements increase $CO_2$ emissions. Population density mainly affects urban $CO_2$ emissions through indirect effects, and every 1% increase in population density in neighboring cities will reduce $CO_2$ emissions by 0.229%. The elasticity coefficients of direct and indirect effects of the technical level are 0.027 and 0.072, respectively, which both pass the 1% significant-level test. In addition, FDI, GI and PT exert a remarkable negative effect on the discharge of $CO_2$; PGDP, EI and TC exhibit a significant positive effect on the discharge of $CO_2$. There is an insignificantly negative association between $CO_2$ discharge and IS, which may suggest a bidirectional relationship between IS and $CO_2$ emissions.

**Table 5.** Effects of comprehensive urbanization indices and meteorological factors on carbon emissions from 2000 to 2018.

| Variables | Wx | Direct | Indirect | Total | Wx | Direct | Indirect | Total |
|---|---|---|---|---|---|---|---|---|
| CU | −0.072 *** | 0.115 *** | 0.041 | 0.156 *** | −0.062 *** | 0.116 *** | 0.070 | 0.186 *** |
| | (−4.278) | (11.439) | (0.750) | (2.609) | (−3.663) | (11.970) | (1.290) | (3.217) |
| lnPD | −0.102 *** | 0.007 | −0.309 ** | −0.302 ** | −0.081 *** | 0.011 | −0.229 ** | −0.218 * |
| | (−3.388) | (0.362) | (−2.509) | (−2.259) | (−2.693) | (0.607) | (−2.002) | (−1.747) |
| lnPGDP | −0.018 | 0.109 *** | 0.202 *** | 0.312 *** | −0.018 | 0.108 *** | 0.197 *** | 0.304 *** |
| | (−0.881) | (11.427) | (3.044) | (4.484) | (−0.902) | (10.956) | (2.894) | (4.236) |
| lnTL | 0.002 | 0.026 *** | 0.071 *** | 0.098 *** | 0.003 | 0.027 *** | 0.072 *** | 0.099 *** |
| | (0.364) | (9.629) | (4.056) | (5.233) | (0.625) | (9.959) | (4.268) | (5.545) |
| lnIS | −0.118 *** | 0.054 *** | −0.265 *** | −0.212 ** | −0.092 *** | 0.048 *** | −0.181 * | −0.133 |
| | (−4.250) | (3.191) | (−2.714) | (−1.994) | (−3.278) | (2.879) | (−1.860) | (−1.276) |
| lnFDI | −0.012 *** | 0.006 *** | −0.026 ** | −0.020* | −0.012 *** | 0.005 *** | −0.025 ** | −0.019 * |
| | (−4.439) | (3.594) | (−2.520) | (−1.816) | (−4.249) | (3.536) | (−2.446) | (−1.789) |
| lnEI | 0.020 ** | 0.040 *** | 0.167 *** | 0.206 *** | 0.022 ** | 0.040 *** | 0.161 *** | 0.201 *** |
| | (2.286) | (8.555) | (4.922) | (5.599) | (2.448) | (9.058) | (4.993) | (5.787) |
| lnGI | −0.036 *** | 0.017 *** | −0.077 ** | −0.061 | −0.036 *** | 0.016 *** | −0.077 ** | −0.061 * |
| | (−3.683) | (3.524) | (−2.139) | (−1.577) | (−3.771) | (3.416) | (−2.390) | (−1.779) |
| lnPT | −0.030 *** | 0.009 * | −0.078 * | −0.069 | −0.034 *** | 0.006 | −0.094 ** | −0.088 ** |
| | (−2.754) | (1.679) | (−1.830) | (−1.490) | (−3.115) | (1.159) | (−2.289) | (−1.973) |
| lnTC | 0.025 ** | 0.022 *** | 0.138 *** | 0.160 *** | 0.025 ** | 0.022 *** | 0.131 *** | 0.154 *** |
| | (2.059) | (3.594) | (2.795) | (2.996) | (2.058) | (3.838) | (3.002) | (3.253) |
| lnPREC | | | | | −0.008 | 0.030 ** | 0.044 | 0.075 * |
| | | | | | (−0.447) | (2.394) | (1.061) | (1.819) |
| lnSARD | | | | | 0.195 *** | −0.054 | 0.475 *** | 0.420 *** |
| | | | | | (2.617) | (−0.923) | (3.015) | (2.962) |
| lnTEMP | | | | | 10.918 *** | −5.698 *** | 21.632 *** | 15.935 *** |
| | | | | | (4.850) | (−3.464) | (4.051) | (2.955) |
| lnWIND | | | | | 0.020 | 0.051 *** | 0.181 *** | 0.233 *** |
| | | | | | (0.881) | (3.860) | (2.954) | (3.712) |
| rho | | | | 0.746 *** | | | | 0.736 *** |
| | | | | (69.437) | | | | (65.840) |
| sigma2_e | | | | 0.009 *** | | | | 0.009 *** |
| | | | | (50.730) | | | | (50.726) |
| Observations | | | | 5396 | | | | 5396 |
| R-squared | | | | 0.851 | | | | 0.860 |
| Number | | | | 284 | | | | 284 |

Note: z-statistics in parentheses. *** $p < 0.01$, ** $p < 0.05$, * $p < 0.1$.

## 4. Discussion

### 4.1. Differences in the Spatial Distribution of $CO_2$ Emissions

The inconsistency in the standards and the availability of statistical data has brought enormous difficulties to the calculation of $CO_2$ emissions at the urban scale. In this study, we used NTL data to estimate urban $CO_2$ emissions in China, providing effective support for the assessment of $CO_2$ emissions in Chinese cities. The distribution of $CO_2$ emissions shows significant spatial differences (Figure 2), which may be related to the development patterns [7]. As a former industrial base in China, Northeast China also has a significant amount from emissions with $CO_2$. The high level of discharge of $CO_2$ in northern China is closely related to its coal resources [48], which cause a high discharge of $CO_2$ in northeastern China. Not only is the structure of the industry skewed towards heavy industry, but the dependence on coal resources for energy can also be transformed into a base for export of high-carbon products. As an established urban spatial pattern, urban agglomeration promotes not only regional economic growth but also the spatial connection of emissions of carbon distribution on the urban level [49]. For instance, the BTH urban agglomeration, including the capital, is the most typical, one which, as the population, economic and political center of China, is surrounded by many energy-intensive enterprises which bring high levels of $CO_2$ emissions. As for coastal cities, they are usually marked by rapid urbanization, a large population, developed heavy industry, and, frequently, an import and export trade because of their superior geographical location, particularly in the YRD and

PRD. Although most cities situated in central China are experiencing a slow growth in $CO_2$ emissions, they have still experienced rapid urbanization (e.g., Chengdu–Chongqing). The development of western China, which contains 70% of country's land area and approximately 30% of the population, is relatively low, and its $CO_2$ emissions are also low. These spatial features are going to have some implications for the distribution of emissions from $CO_2$ in China.

Furthermore, over the past 19 years, the discharge of carbon can be roughly grouped into four phases: rapid growth period (2000–2004, the average annual growth rate = 9.95%); slow growth period (2004–2008, the average annual growth rate = 4.93%); rapid growth period (2008–2013, the average annual growth rate = 8.92%); and slow growth period (2014–2018, the average annual growth rate = 4.99%). The spatial difference in China's $CO_2$ emission growth rate is owing to imbalanced socioeconomic development levels across China [15]. We additionally observe a decrement in $CO_2$ emissions in 2014, which was the first reduction in China's carbon emissions since 1999, as stated by the International Energy Agency, thus contributing to the discharge reduction of global $CO_2$. Nevertheless, the delinking between the development of the economy and the discharge of $CO_2$ has still not been achieved [50]. The consumption of energy remains the main driver of the discharge of $CO_2$ [51] because of the dominant position of coal, given the limitations of renewable energy [52]. Hence, China is more likely to be in a phase of growth in carbon emissions in the coming years, and reducing $CO_2$ emissions is the key to achieving the dual carbon goal.

### 4.2. Explanation of the Differences in the Influences of Driving Factors of $CO_2$ Emission

Regional $CO_2$ emissions have significant spatial correlations, and ignoring the impacts of the correlations can lead to biased results. This study fully considers the driving factors of $CO_2$ emissions and uses spatial econometric models to study the direct effects and spillover effects of urbanization on $CO_2$ emissions, and to further understand the relationship between urbanization and $CO_2$ emissions. According to our results, urbanization has a multifaceted influence on the change of China's discharge of $CO_2$. Specifically, the degrees of influence of PU and EU on $CO_2$ emission are relatively low due to their inconsistent spatial patterns [53]. However, both variables significantly promote local carbon emissions, which may be associated with the increase of fossil fuel usage in times of rapidly growing populations and economies [54]. With regard to population urbanization, on one side, substantial urban–rural migration increases energy consumption for residence and transportation, worsening the discharge of $CO_2$ [55]. On the other side, PU promotes the transition to clean fuel types by side effects, suppressing carbon emissions to some extent [53,56]. In addition, China's strict household registration system and immature migrant worker welfare system, combined with social environment constraints, limit the flow of people moving to cities from rural areas, thus hindering the China's overall process of population urbanization [57]. Consequently, the effect of PU on the discharge of $CO_2$ is offset. Because coal continues to dominate the energy usage pattern in China [58], EU is instrumental in increasing local emissions of $CO_2$. However, there are two-way causative interactions between economic structure and the discharge of $CO_2$ [17]. Therefore, the effect of the EU also becomes insignificant. Compared with PU and EU, LU seems to have a substantial influence on $CO_2$ emissions. As land urbanization typically occurs after the expansion of construction sites, construction site emissions and road traffic emissions further increase urban carbon emissions [59]. Moreover, the reduced vegetation area and increased urban heat-island-effect also play a negative role in the dispersion of atmospheric pollutants. Here, LU is identified as the main indicator representing urbanization's effect on the discharge of $CO_2$, which is supported by our regression model. Additionally, the comprehensive urbanization index also confirms this view. This is probably attributable to the change in land use resulting from the expansion of built-up areas, which not only increases $CO_2$ emissions but also reduces carbon sinks [60].

In terms of the control variables, a rise in PGDP will increase $CO_2$ emissions in local and adjacent cities. Cities with higher economic levels tend to have higher consumption,

leading to an increase in output. Meanwhile, it will also boost the economic development in surrounding cities, forcing the growth in the discharge of $CO_2$. Similarly, EI obviously increases $CO_2$ emissions. In addition, traffic congestion is also a significant positive driver. This may be due to the reduced car fuel economy caused by traffic congestion, further contributing to energy waste and the discharge of carbon. Road traffic is the key contributor to the emissions of $CO_2$ leading to climate change. Then, there is a trifling negatively correlated trend between FDI and the discharge of $CO_2$, with a strong positive direct influence and a negative indirect influence. On the one side, to avoid costly environmental compliance, some pollution-intensive corporations always engage with countries with weak environmental legislation; thus, FDI may lead to higher pollution levels [61]. On the other hand, FDI promotes technological innovation and reduces energy demand, and its demonstration effect reduces $CO_2$ emissions from neighboring cities [30,62]. Furthermore, government intervention can maximize social welfare, which is essential to addressing environmental challenges and the usage of energy [32,33]; therefore, GI does have a considerable negative effect on $CO_2$ emissions. Moreover, PT also reduces $CO_2$ emissions, which can be attributed to the reduction in the usage of private automobiles [40]. Public transport has a certain potential for promoting emission reduction and is an important part of sustainable urban planning. Notably, unlike previous findings, our results disclose a strong negative connection between population density and the discharge of carbon, which could be linked to the stronger spillover effects of technology and knowledge sharing caused by population agglomeration [63]. Although technological innovation is one of the most successful methods of implementing energy conservation and development of sustainability in China, there are still many uncertainties in energy retrenchment and the discharge reduction of $CO_2$ attainable due to increased perfection in traditional factors of productivity [34].

Apart from urbanization and socio-economic factors, meteorological factors also have an important influence on $CO_2$ emissions. The meteorological factors involved in this study contribute to some extent to the augmented discharge of $CO_2$. This may be put down to changes in meteorological conditions that increase people's various efforts to address their risks, thereby increasing energy consumption, and unstable meteorological conditions also have a certain inhibitory effect on energy utilization efficiency. Electricity consumption reaches its peak in summer because of the dramatic increase in urban cooling energy consumption [64]. Moreover, under moist heat conditions, building energy consumption is more responsive to changes in temperature and humidity [65]. In addition, clean energy, the major means of energy economization and discharge abatement in China, is also sensitive to meteorological factors [66]. Rises in wind speed, humidity and temperature increase the electricity consumption of solar and wind energy, which may promote carbon emissions.

*4.3. Limitations and Future Directions*

Although this essay has provided a connection between the discharge of $CO_2$ and urbanization, several limitations remain. First, owing to the inconsistency of data between different sensors and the low spatial resolution of NTL data, there is uncertainty in using NTL data to estimate $CO_2$ emissions, an uncertainty which may lead to inaccurate estimation. In the future, the technical methods of $CO_2$ emission accounting should be further improved to enhance the applicability of the data. Second, although the spatial regression model was employed for quantifying the spillover effect of urbanization on the discharge of carbon, the spillover effect we measured is an average effect, which cannot represent the individual effect of each geographical unit. Finally, the unevenness of development in China's regions led to a spatial disparity of the urbanization impact as regards the discharge of carbon. In future, we are open to considering the use of other methods to discuss urbanization's effects on mechanisms for the discharge of carbon in more detail.

## 5. Conclusions

In social development and climatic variation, cities are vital. Prior work suggested that urbanization alters the discharge of carbon; nevertheless, the control of meteorological factors and urbanization's spillover effects to the discharge of carbon within the adjacent region still lacks quantification. Also, different urbanization courses, like population, land and economic urbanization, should be delineated. Here, we used SDM for quantifying the direct impact and spillover impact from urbanization to energy-related carbon discharge for 284 cities in China by estimating urban-level energy-related $CO_2$ emissions from remotely sensed nighttime light data. Our results indicate that China's $CO_2$ emissions have significant spatial autocorrelation. Urbanization in China has significantly increased $CO_2$ emissions, with demographic and economic urbanization making a strong impress on local discharge of carbon. As for LU, it has a stronger promoting effect on $CO_2$ emissions than do PU or EU, especially for the spillover effect. Specifically, a city's discharge of carbon increases 0.066% for each 1% rise of its own urbanization level, and 0.492% with each 1% increase in the mean urbanization standard of its neighbors. It is worth pointing out that the spillover effect of population agglomeration contributes more strongly to carbon reduction. Based on the improvement of traditional technologies and means, carbon emissions may be reduced. The results also point out that PGDP, EI and TC are the main factors driving $CO_2$ emissions, while foreign direct investment, public transport and government intervention will significantly decrease $CO_2$ emissions. In addition, meteorological factors have a substantial impact on $CO_2$ emissions by affecting clean energy use in China.

Considering the consequences, several alterations to policies are suggested. First, when formulating $CO_2$ emission reduction policies, the spatial correlation and heterogeneity of $CO_2$ emissions should be considered. Through the construction of a regionallyintegrated governance platform, we can promote the exchange of information and policy interaction between cities. As for the carbon emission hot cities in the eastern coastal areas (such as BTH and YRD), they have a high level of urbanization, and extra care should be taken to protect the ecological environment in their future urbanization processes. As an example, one could formulate the overall regional urban planning and layout, optimize the deployment of resource elements and play out the scenario of economic agglomeration. The quality of urban development in the western and central districts ought to be improved to prevent urban unordered sprawl. Second, the government ought to establish a scientific land planning system because of the intimate mutual relation between land urbanization and the discharge of carbon. The speedy sprawl of built-up areas must be controlled by fostering sustainable growth on urban land. The conversion of natural land into construction land should be restricted, because green space plays an important role as a carbon sink. Third, GDP per capita and the intension of energy are the main elements promoting the discharge of $CO_2$. For the way forward, the government should reasonably upgrade the economic development pattern, appropriately improve the percentage of tertiary industry, reduce the reliance on the fossil bunkers, vigorously pursue the expansion of renewable energy sources and actively foster a low-carbon economy. The spillover effects of population agglomeration and FDI have an arresting negative effect on $CO_2$ discharge. For future urban development, it is auspicious and important to raise the environmental consciousness of residents and enhance the introduction of high-quality talents and foreign capital. In addition, traffic conditions also have a significant bearing on the discharge of $CO_2$, one which would be lessened by restricting the use of private cars, developing public traffic infrastructure and reducing stress on road traffic.

**Supplementary Materials:** The following supporting information can be downloaded at: https://www.mdpi.com/article/10.3390/rs15164093/s1.

**Author Contributions:** Methodology, L.F.; Validation, M.Z.; Formal analysis, X.M.; Resources, X.G.; Data curation, X.W.; Writing—original draft, L.S.; Writing—review & editing, X.M. and L.F.; Supervision, L.F. All authors have read and agreed to the published version of the manuscript.

**Funding:** This work was financially supported by the National Natural Science Foundation of China (41975044, 41925007, 42101385, and 41801021).

**Data Availability Statement:** Not applicable.

**Conflicts of Interest:** No potential conflict of interest was reported by the authors.

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
