# Peer review of "Investigating the Direct and Spillover Effects of Urbanization on Energy-Related Carbon Dioxide Emissions in China Using Nighttime Light Data"

_remotesensing, doi:10.3390/rs15164093_

Round 1
Reviewer 1 Report
I believe that this study on the quantification of the urban CO2 emissions in China from 2000 to 2018 in relation with socio-economic and climate variable is of great importance and topicality. China is the world's largest emitter of CO2 and therefore it is crucial that it applies reduction policies if it is to somehow mitigate global climate change.
The use of night lights to determine emissions is a frequently used technique, but the authors correlate it with several factors and calculate both the temporal and spatial viariation of China's emissions for 284 cities.
The methodology is well explained and repeteable and the results are supported by statistical index. The limitations of the study are well reported and the conclusions with suggestions to mitigate the heavy use of fossil fuels can help policy makers. The importance of working across cities and thus nationwide is evident from the spatial analysis performed. My opinion is that this work is definitely ready in present form for publication.
Check only some typos present in the text.
Author Response
Thanks very much, we have checked and revised the whole manuscript.
Reviewer 2 Report
In eq. 1, CO2 emission coefficient is used, whether it is reasonable to use 44/12 as the coefficient?
How to deal with multi-source data with different spatio-temporal resolutions?
The research scope of Fig.2 and Fig. 5, Fig. 6 is different. Please explain what it means.
The selection of influencing factors on carbon emissions is arbitrary. Whether variables are selected repeatedly, such as energy intensity and technology level variables.
The paper lacks robustness test to prove the existing conclusions.
Minor editing of English language required
Author Response
Comment1:In eq. 1, CO2 emission coefficient is used, whether it is reasonable to use 44/12 as the coefficient?
Response 1:Thank you very much, we have modified it. Ki represents the carbon emission factor for energy of category i. It needs to be multiplied by the molar ratio of CO2 to C to convert to CO2 emissions. And our settlement results are highly consistent with those of China Emission Accounts and Datasets (CEADs).
Comment 2:How to deal with multi-source data with different spatio-temporal resolutions?
Response 2:Thanks, we select the data from 2000 to 2018 according to the availability of the data, and resample the multi-source remote sensing data to 1 km resolution for regional statistics.
Comment 3:The research scope of Fig.2 and Fig. 5, Fig. 6 is different. Please explain what it means.
Response 3:Thank you very much.We have explained it in manuscript.
Comment 4:The selection of influencing factors on carbon emissions is arbitrary. Whether variables are selected repeatedly, such as energy intensity and technology level variables.
Response 4:Thank you very much. According to previous studies, in order to avoid the deviation caused by omitting variables, we have fully considered the factors affecting carbon emissions. Moreover, we test the correlation between all variables and screen the influencing factors to avoid the correlation between variables affecting the accuracy of the results.
We use research and technology expenditure as a percentage of total financial expenditure to express the level of technology, which highlight China's impact on carbon emissions in terms of scientific and technological innovation. The results show that the level of technology will significantly increase city-level carbon emissions. This further show that China's current low-carbon technology progress is not enough, more reflect in improving the efficiency of traditional processes. Moreover, the results of multicollinearity test also show that there is no correlation between technical level and energy intensity.
Comment 6:The paper lacks robustness test to prove the existing conclusions.
Response 6: Thanks for your valuable suggestion. First of all, we use the method of multiple linear regression to study the influencing factors of carbon emissions. Secondly, the results of spatial regression model showedhigher goodness of fit. Finally, we replace the variables and calculated the comprehensive urbanization index to further prove the credibility of the results.
Reviewer 3 Report
This paper distinguish the influence that various urbanization factors have on emissions of CO2. The finding and results is very interesting to readers. The paper is well written. I suggested minor revision that requires authors to clarify a few unclear points.
1. Table 1 doesn't give the CO2 emission factor of electricity. Has this part of emission been included in the accounting?
2. Please explain the meaning of each column in Table 2, such as "N", "mean", etc. Are these results the national average from 2000-2018?
3. In eq. 3, the meaning of j is not provided, may be same as i? if so, do the i and j need to be adjacent? Please explain the meaning of "territory's observation value". Under the purpose of this research, how to explain I=0?
4. This paper analyzed the influence of various factors on emissions. But is the correlation between each factor taken into account? How to remove autocorrelation among these factors?
5. How are various urbanization indicators determined?
6. This paper studies the influence of meteorological factors on carbon emissions, but there is little exposition on this part.
7. The key conclusions of this paper are not discussed enough. I hope the author can give some supplement.
8. What is the limitation and uncertainty that exists in the result?
Minor editing of English language required
Author Response
- Table 1 doesn't give the CO2 emission factor of electricity. Has this part of emission been included in the accounting?
Response: Thanks for your careful review. As the electricity emission factor is different in different regions of China. In this study, we approximately take the average value of 1 as the carbon emission factor of electricity.
- Please explain the meaning of each column in Table 2, such as "N", "mean", etc. Are these results the national average from 2000-2018?
Response: Thank you very much. We have added a corresponding explanation in Table 2.
- In eq. 3, the meaning of j is not provided, may be same as i? if so, do the i and j need to be adjacent? Please explain the meaning of "territory's observation value". Under the purpose of this research, how to explain I=0?
Response: Thanks for your careful review, the expression here may not be clear, but we have modified it.
The j represents a city that is different from i, and i and j do not need to be adjacent to each other, and the degree of spatial connection between city i and j is expressed according to the different weights given to it by the spatial weight matrix.
- This paper analyzed the influence of various factors on emissions. But is the correlation between each factor taken into account? How to remove autocorrelation among these factors?
Response: Thanks for your valuable suggestion.We have used the variance inflation factor (VIF) to test the correlation between variables. These variables are selected to be included in our study on the premise of fully considering the influencing factors of carbon emissions. The VIF values of all variables are less than 5, indicating that there is no multicollinearity between variables.
- How are various urbanization indicators determined?
Response: Thanks, According to the previous research, we quantify urbanization from three aspects: population, economy and land, which have been explained in the manuscript.
- This paper studies the influence of meteorological factors on carbon emissions, but there is little exposition on this part.
Response:Thank you. Because the meteorological condition of a city is relatively stable, and it affects CO2 emissions mainly by affecting energy use efficiency, it is not the main factor of this study.
- The key conclusions of this paper are not discussed enough. I hope the author can give some supplement.
Response: Thanks for your constructive suggestion. We have added and modified some discussions.
- What is the limitation and uncertainty that exists in the result?
Response: Thank you very much. First, there is uncertain to estimate CO2 emissions using the linear relationship between NTL and carbon emissions. Second, although the spatial regression model is employing for quantifying the spillover effect of urbanization on the discharge of carbon, the spillover effect we measured is an average effect, which cannot represent the individual effect of each geographical unit.
Round 2
Reviewer 2 Report
Accepted